# Mpox: Clinical Outcomes and Impact of Vaccination in People with and without HIV: A Population-Wide Study

**DOI:** 10.3390/microorganisms11112701

**Published:** 2023-11-03

**Authors:** Raquel Martín-Iguacel, Carles Pericas, Andreu Bruguera, Gemma Rosell, Erica Martínez, Yesika Díaz, Lucia Alonso, Daniel Kwakye Nomah, Jose Luis Blanco, Pere Domingo, Patricia Álvarez-López, Maria Saumoy Linares, Consuelo Vilades Laborda, Arantxa Mera, Sonia Calzado Isbert, Isik Somuncu Johansen, José M. Miró, Jordi Casabona, Josep M. Llibre

**Affiliations:** 1Centre of Epidemiological Studies of HIV/AIDS and STI of Catalonia (CEEISCAT), Health Department, Generalitat de Catalunya, 08916 Badalona, Spain; abruguerar@iconcologia.net (A.B.); ydiazr@iconcologia.net (Y.D.); lalonso@lluita.org (L.A.); dnomah@igtp.cat (D.K.N.); jcasabona@iconcologia.net (J.C.); 2Department of Infectious Diseases, Odense University Hospital, 5000 Odense, Denmark; isik.somuncu.johansen@rsyd.dk; 3Epidemiology Service, Public Health Agency of Barcelona, 08023 Barcelona, Spain; cpericas@aspb.cat; 4Department of Medicine, University of Barcelona, 08036 Barcelona, Spain; 5Institute of Biomedical Research Hospital de la Santa Creu i Sant Pau (IIB Sant Pau), 08041 Barcelona, Spain; ericamartinez@gencat.cat; 6CIBER Epidemiologia y Salud Pública (CIBERESP), 28029 Madrid, Spain; 7Department of Paediatrics, Obstetrics and Gynecology and Preventive Medicine, Universitat Autònoma de Barcelona, 08916 Badalona, Spain; 8Subdirecció General de Vigilància i Resposta a Alertes i Emergències de Salut Pública—ASPCAT, Teià, Cataluña, Spain; gemma.roselld@gencat.cat; 9Fight Infections Foundation, Badalona, 08916 Barcelona, Spain; 10Fundació Institut D’investigació en Ciències de la Salut Germans Trias I Pujol (IGTP), 08916 Badalona, Spain; 11Hospital Clínic-Institut d’Investigacions Biomèdiques August Pi i Sunyer, University of Barcelona, 08916 Barcelona, Spain; jlblanco@clinic.cat (J.L.B.);; 12Infectious Diseases Unit, Hospital Universitari de la Santa Creu i Sant Pau, 08916 Barcelona, Spain; pdomingo@santpau.cat; 13Department of Infectious Diseases, Hospital Universitari de la Vall d’Hebron, 08916 Barcelona, Spain; patricia.alvarez@vallhebron.cat; 14HIV and STD Unit, Infectious Disease Department, Hospital Universitari de Bellvitge, Hospitalet de Llobregat, 08916 Barcelona, Spain; msaumoy@bellvitgehospital.cat; 15Department of Internal Medicine, Hospital Universitari Tarragona Juan XXIII, Universitat Rovira i Virgili, Tarragona, Spain; cvilades@gmail.com; 16Centro de Investigación Biomédica en Red de Enfermedades Infecciosas (CIBERINFEC), Instituto de Salud Carlos III, Madrid, Spain; 17Department of Internal Medicine, Hospital de Palamós, 17230 Girona, Spain; arantxameraf@gmail.com; 18Department of Infectious Diseases, Parc Taulí University Hospital, Sabadell, 08916 Barcelona, Spain; scalzado@tauli.cat; 19Infectious Diseases Department, University Hospital Germans Trias i Pujol, Badalona, 08916 Barcelona, Spain; jmllibre@lluita.org

**Keywords:** mpox, sexually transmitted infections, HIV, immunosuppression

## Abstract

We investigated differences in mpox clinical outcomes in people with HIV (PWH) and without HIV (PWoH) and the impact of vaccination in Catalonia, Spain. We used surveillance data and the PISCIS HIV cohort. We included all confirmed mpox cases (May–December 2022). Of 2122 mpox cases, the majority had mild disease, 56% were Spanish, and 24% were from Latin America. A total of 40% were PWH, with a median CD4+T-cell of 715 cells/μL; 83% had HIV-RNA < 50 copies/mL; and 1.8% CD4+T-cell < 200 cells/μL. PWH had no increased risk for complications, except those with CD4+T-cell < 200 cells/μL. PWH with CD4+T-cell < 200 cells/μL were more likely to be from Latin America, had more generalized exanthema, and required hospitalization more frequently (*p* = 0.001). Diagnosis of other sexually transmitted infections (STIs) was common, both at mpox diagnosis (17%) and two years before (43%). Dose-sparing smallpox intradermal vaccination was accompanied by a sharp decrease in mpox incidence in both populations (*p* < 0.0001). In conclusion, unless immunosuppressed, PWH were not at increased risk of severe disease or hospitalization. Mpox is a marker of high-risk sexual behavior and was associated with high HIV and STI rates, supporting the need for screening in all mpox cases. Ethnicity disparities demonstrate the need for interventions to ensure equitable healthcare access. Dose-sparing smallpox vaccination retained effectiveness.

## 1. Introduction

The ongoing first global outbreak of the mpox virus in humans (formerly known as monkeypox) was first recognized in May 2022 when several countries in Europe and North America, where the disease was not endemic, declared the first mpox cases [1,2,3,4,5,6]. The World Health Organization declared the infection a public health emergency of international concern on 23 July 2022 [7]. As of 19 September 2023, over 90,000 laboratory-confirmed cases have been reported worldwide from 115 Member States across all six WHO regions, with over 26,000 cases in the European region [8].

Mpox is a zoonotic disease caused by a double-stranded DNA virus classified in the Orthopoxvirus genus of the Poxviridae family. The disease is endemic to western and central Africa, especially in the Democratic Republic of Congo. Transmission before 2022 was predominantly zoonotic, related to direct contact with infected animals. Most cases detected outside the endemic areas responded to sporadic and limited outbreaks linked to exotic pet trade or travel activity [9]. During the current global outbreak, the transmission has occurred through close person-to-person contact, predominantly sexual contact through high-risk sexual practices, without epidemiological links to western or central Africa, and mainly, but not exclusively, in men who have sex with men (MSM) [8]. HIV infection has been overrepresented in the current outbreak, with approximately 40–50% of the cases being coinfected with HIV [10]. The illness is usually mild symptomatic, including few skin and/or mucosal lesions at inoculation sites, fever, and local lymphadenopathy, and 15–30% present with proctitis [10,11]. However, more severe cases have been described, especially in immunosuppressed patients, e.g., HIV-related immunosuppression, with necrotizing skin lesions, lung involvement, central nervous system infection, secondary bacterial infections, sepsis, and ocular involvement [12]. As of 19 September 2023, a total of 157 deaths have been reported globally, which demonstrates that mpox can be a life-threatening disease in an immunosuppressed host and that ongoing measures are needed to prevent infection and disease spread [8].

The earlier vaccinia virus vaccination administered against smallpox provided cross-immunity to the mpox virus as both viruses are closely related to Orthopoxviruses [13,14]. Discontinuation of routine smallpox vaccination following smallpox eradication in the early 1980s and thus waning immunity with no immunity coverage in the population <40 years of age has probably been an important risk factor for the current emergence of mpox as a public health thread [9].

The objective of this study is to describe the epidemiological data collected during the outbreak in Catalonia in 2022 as part of the public health surveillance program with a special focus on the differences in clinical outcomes at a population level between people with HIV (PWH) and without HIV (PWoH) and in those with severe immunosuppression among PWH. Furthermore, we aim to explore the effectiveness of dose-sparing smallpox vaccination, implemented at the beginning of the outbreak due to restrained vaccine availability, in controlling the outbreak.

## 2. Materials and Methods

### 2.1. Setting

On 1 January 2023, Catalonia had a population of 7.7 million citizens and an estimated HIV prevalence among the adult population of 0.4%. The Catalan healthcare system provides universal, tax-funded healthcare and antiretroviral therapy (ART); mpox tests; mpox vaccination; and mpox treatment are provided free of charge to all citizens.

Mpox is a nationally notifiable disease in Spain, and mpox cases are notified by sexually transmitted infections (STIs) clinics, primary healthcare, and hospitals to the Surveillance System of the Regional Ministry of Health [15]. Data are collected using standardized case report forms (CRF).

### 2.2. Study Design and Data Sources

This is a surveillance study conducted in Catalonia, Spain, including all confirmed cases of mpox between 6 May and 19 December 2022. A confirmed mpox case was defined as a positive mpox virus polymerase chain reaction (PCR) laboratory result. Specimens were collected from suspected lesions for each suspected case. Laboratory analyses were conducted in the National Reference Laboratory in Madrid until 31 May 2022. From this date, the mpox testing capacity was enhanced and was available in 12 different laboratories in Catalonia.

Data collection was performed by trained epidemiologists who completed the mpox standardized CRFs. We linked this surveillance data with the Catalan HIV PISCIS cohort (Catalonian and Balearic Islands HIV cohort) and PADRIS (Public Data Analysis for Health Research and Innovation Program).

The study design of PISCIS is described elsewhere [16]. PISCIS is an ongoing, prospective, multicentre, population-based cohort from 1998 that includes all PWH aged ≥16 years followed in one of the 16 collaborating hospitals in Catalonia, representing 84% of all diagnosed PWH. Data are updated yearly and include demographics, date of HIV diagnosis, AIDS-defining events, ART, and measurement of CD4+T-cell count and plasma HIV-RNA over time.

PADRIS is a research-oriented big data repository that gathers and cross-matches real-world health data generated by Catalonia’s public health care system (SISCAT). Programmatic Health data are provided by the Catalan Agency for Health Quality and Evaluation (AQuAS). The registry includes individual-level comorbidity data from hospital discharge diagnoses and primary health care from 2005 according to the International Classification of Disease 10th revision (ICD-10), laboratory, microbiology, prescription, and epidemiological surveillance data from mandatory notification systems of infectious diseases [17]. From PADRIS, we retrieved information for other notifiable sexually transmitted infections (STIs) and comorbidity data using ICD-10 codes, defining comorbidities according to the Swedish National Study of Aging and Care in Kungsholmen (SNAC-K) cohort (www.snac-k.se accessed on 30 October 2023) (Appendix A).

Charlson comorbidity scores at baseline were calculated in all groups using diagnosis coded in PADRIS, excluding AIDS-defining event, which was assessed separately [18].

### 2.3. Study Population

We included all individuals with laboratory-confirmed mpox between 6 May and 19 December 2022.

## 3. Statistical Analysis

Continuous variables were described as the median and the interquartile range (IQR), whereas categorical variables were presented as the frequency and percentage of available data. We used the chi-squared, T, and Mann–Whitney U tests to compare variables between the groups.

We conducted descriptive statistics on demographics and clinical outcomes comparing PWoH and PWH individuals. We also described the prevalence of other concurrent STIs at mpox diagnosis (with a time window of 8 weeks on either side of mpox diagnosis) and in the two years before mpox.

Descriptive statistics were also conducted in the group of PWH, comparing those with CD4+T-cell counts < 200 and ≥200 cells/μL at mpox diagnosis.

We constructed an epidemiological curve with the cumulative confirmed cases of mpox per week to explore the temporal relation with the initiation of dose-sparing smallpox vaccination in Catalonia. We used negative binomial regression to assess the impact of vaccination on the daily counts of mpox cases and provided a *p*-value as a significance test.

We used STATA software for the analysis (v.16; Stata Corp, College Station, TX, USA).

## 4. Ethics Approval

The PISCIS cohort has received ethical approval from Germans Trias i Pujol University Hospital’s Clinical Research Ethics Committee, reference number EO-11-108, and patient data extraction is allowed by the 203/2015 Decree from the Catalan Health Department. All data are pseudo-anonymized by regulation 2016/679 of the European Parliament.

## 5. Results

A total of 2122 individuals were reported with confirmed mpox during the study period. Of these, 842 (39.7%) had HIV-coinfection (Figure 1). Overall, the most probable reported transmission mode of mpox was sexual (88.2%) in men (97.6%), with a median age at mpox diagnosis of 38 years (IQR: 32–44). A total of 59% were of Spanish origin, 21% were from Latin America, 13% were from other European countries, and the rest were from different countries worldwide. Approximately 9% had traveled internationally three weeks before mpox diagnosis.

The clinical characteristics of PWoH and PWH are shown in Table 1. Most individuals were symptomatic with a mild and self-limiting disease. The symptoms were similar in the PWoH and PWH groups, except the latter had more frequent fever and generalized exanthema (*p* = 0.015 and 0.002, respectively). The rash was mainly described as maculopapular, vesicular, umbilical, or pustular, and the latter two were more commonly described among PWH (*p* = 0.011). The complication rate (bacterial skin infections, pneumonia, cornea infection, proctitis, and/or hospitalization requirement) was low overall (5.0%), without significant differences in both groups (*p* = 0.13). Approximately 2% of the patients developed bacterial superinfection of the skin lesion, mainly in the anogenital region, followed by the facial area, and 1.4% developed proctitis. Only two patients in the PWH group developed pneumonia and three corneal infections. Forty-four (2.1%) individuals required hospital admission without differences in the two groups (*p* = 0.16). No severe complications (sepsis, encephalitis, myocarditis) were observed except one unvaccinated PWoH who required admission to the intensive care unit (ICU). There were no mpox-related deaths in this series.

Overall, 16.5% of mpox cases were diagnosed with a concurrent notifiable STI at mpox diagnosis, and 43% had had at least one notifiable STI in the two years before mpox compared to 8.5% of PWH without mpox (*p* < 0.0001) (Figure 2, Appendix A).

PWH developing mpox were mainly men (99.7%) and MSM (92.3%) with a median age of 41 years (IQR 35–46). The median CD4+T-cell count at mpox diagnosis was 715 cells/μL (IQR 526–909), with only 12 (1.8%) individuals with CD4+T-cell count < 200 cells/μL. A total of 83% of PWH had plasma HIV-RNA < 50 copies/mL at mpox diagnosis, with 66.7% in the group of individuals with CD4+T-cell count < 200 cells/μL (*p* = 0.12). There were no cases of concurrent mpox and HIV diagnosis in the PISCIS/PADRIS database. However, we cannot rule out concurrent new HIV diagnosis at mpox diagnosis in the remaining 169 PWH who were not included in PISCIS since only the HIV status was recorded in the standardized CRFs for these patients.

We compared the clinical presentation of mpox in the 673 PWH included in the PISCIS cohort with CD4+T-cell count < 200 versus ≥ 200 cells/μL (Table 2). The median CD4+T-cell count was 122 cells/μL (IQR 72–176) and 724 cells/μL (IQR 540–912), respectively. The Charlson comorbidity scores were similar in both groups, with more than 3/4 having an index of 0. However, malignancy and earlier AIDS-defining events were more common in PWH with CD4+T-cell count < 200 cells/μL (25% versus 5.8%, *p* = 0.006, and 33.3% versus 7.7%, *p* = 0.001, respectively). Furthermore, we found that PWH with CD4+T-cell count < 200 cells/μL were more commonly from Latin America (100% versus 41%, *p* < 0.0001), had more generalized exanthema (83.3% versus 44.6%, *p* = 0.008), had more frequent pustular and hemorrhagic exanthema (50.0% versus 20.6%, *p* = 0.013, and 8.3% versus 0.3%, *p* < 0.0001, respectively) and required more frequently hospitalization (16.7% versus 2%, *p* = 0.001). Higher hospitalization rates were also observed in individuals with HIV-RNA ≥ 50 copies/mL compared to HIV-RNA < 50 copies/mL (Appendix A). However, no individual required hospitalization in the ICU, and no severe complications, such as pneumonia, bacterial infection, or proctitis, were recorded.

Smallpox vaccination was introduced in Catalonia on 17 July 2022 and was associated with a statistically significant decline in the mpox incidence, as shown in Figure 3 (*p* < 0.0001).

At mpox diagnosis, 23.7% and 14.5% (*p* < 0.0001) of PWH and PWoH had received at least one fractioned dose of the smallpox vaccine during the outbreak, with a median time from the first dose of the vaccine to the development of the symptoms of 23 days (IQR: 13–48). A higher percentage of PWH (16.5%) compared to PWoH (8.4%) had received smallpox vaccination in childhood (*p* < 0.0001).

## 6. Discussion

We report population surveillance data on 2122 mpox cases notified from 6 May 2022 to 19 December 2022 in Catalonia, in PWoH and PWH. The risk of severe mpox infection or hospitalization was not increased in PWH compared to PWoH, except in immunosuppressed individuals. However, the absolute numbers of immunosuppressed PWH were very low. We found that HIV coinfection or other STIs were common among individuals diagnosed with mpox, which highlights the importance of screening for HIV and STIs in these individuals. We also observed notable disparities in mpox incidence among individuals of Latin American origin. Finally, dose-sparing smallpox intradermal vaccination was implemented at the beginning of the outbreak due to restrained vaccine availability, and it was associated with a significant decline in mpox cases in both groups.

Mpox infection usually had a mild course, and most individuals recovered without complications, both PWH and PWoH. The rate of severe illness was very low, and although both severe disease and death have been described in other reports, we did not observe any severe complications or death cases in this series. In a recent report, the mpox-associated death rate in the US was 1.3 per 1000 cases (and 1.2 per 1000 cases worldwide), mainly observed in immunocompromised unvaccinated individuals [19]. This case fatality rate is lower than reported in outbreaks before 2022 [9].

Transmission during the current mpox outbreak was mainly due to sexual activity in high-risk unprotected sexual practices, mainly in MSM, as evidenced by the numerous previous and concurrent diagnoses of other notifiable STIs in these individuals and the rates of HIV coinfection. Control measures such as contact tracing, information campaigns, and vaccination were pivotal and were introduced early in Catalonia during the outbreak. Vaccination represents a crucial public health measure to reduce disease spread and severity and control the epidemic. The smallpox vaccine (JYNNEOS; Modified Vaccinia Ankara-Bavarian Nordic; MVA-BN) was offered against mpox during the study period in a two-dose regimen. Due to restrained vaccine availability, a dose-sparing vaccination administrered intradermally was initially implemented, fractionating a full-dose vaccine vial into five doses (1 mL) and promoting it to all individuals at high risk. From October 2022, a full second dose was offered to all individuals.

Although the efficacy of the vaccine against mpox has not been assessed in clinical trials, real-world studies have shown good effectiveness [13,14,20,21]. A recent study from Israel evaluating 1037 male individuals who completed only the first subcutaneous dose of the vaccine and were followed up for at least 90 days estimated that the adjusted vaccine effectiveness in preventing infection was 86% (95% confidence interval 59–95%) [22]. A CDC report from December 2022 estimated that the mpox incidence was 7 and 10 times higher among unvaccinated individuals compared to those who had received only one or both doses, respectively [23]. However, in a recent case-control study from the US, authors estimated a lower vaccine effectiveness of 66% and 36% after two and one dose of the vaccine, respectively [20]. 

The smallpox vaccine became available in Catalonia on 17 July 2022. It was recommended for unvaccinated adults with a high risk of contracting the infection, that is, those who engaged in high-risk sexual activities, and for close contacts of confirmed cases who, if they were to contract the disease, had a higher risk of suffering further complications, such as children, pregnant women, and immunocompromised people, if the vaccine could be administrated within four days from exposure to the infection. The introduction of the dose-sparing vaccine was associated with a clear decline in incidence in our population. However, other reasons, like the effect of other public health measures instigated, could have also played a role. A recent study using mathematical modeling, estimated that the dose-sparing intradermal vaccination strategy was moderately effective with better population prevention outcomes in the context of mpox vaccine supply shortages in the acute phase of the pandemic, being able to vaccinate a larger number of people at high risk [24].

Among patients developing mpox, the rates of prior smallpox vaccination were higher in PWH, probably reflecting the recommendations for immunization of high-risk populations at that time and their accessibility through STI and HIV Units. Interestingly, PWH also had higher rates of smallpox vaccination from childhood, possibly related to their older age and the potentially higher smallpox vaccination rates in migrants.

The prevalence of HIV coinfection among mpox cases in our population is similar to that described in the previous case series, at approximately 40% [10,11]. This is most likely due to the overrepresentation of people with high-risk sexual practices, as evidenced by the significant disparities in the prevalence of other STIs in PWH with and without mpox in our study (*p* < 0.0001, Figure 3 and Appendix A). These results highlight the importance of offering HIV testing to all mpox cases and screening for other STIs. Therefore, a mpox diagnosis is an opportunity to prescribe pre-exposure prophylaxis (PrEP), which should be considered in HIV-negative cases to reduce the risk of later HIV infection in this high-risk group. Additionally, initiating ART in PWH who are not yet on ART is essential, especially in cases of advanced HIV infection, to boost immune recovery. Most PWH developing mpox had high CD4+T-cell count at mpox diagnosis and undetectable plasma HIV-RNA, reflecting a population mostly successfully treated for HIV.

Except for HIV-related immunosuppression, PWH had similar mpox clinical outcomes to PWoH. Nevertheless, PWH had more frequent fever and generalized umbilical or pustular exanthema (*p* = 0.015 and 0.002, respectively). Although the absolute numbers of PWH with low CD4+T-cell count < 200 cells/μL and mpox infection were very low (1.8% of all PWH with mpox), we found a more severe illness, with more generalized exanthema, more frequent pustular and hemorrhagic pattern, and with higher rates of hospitalization. One-third of these patients had had a previous AIDS-defining event, and one-fourth had had a malignancy diagnosis, evidencing a population with severe immune suppression and a higher comorbidity burden. However, no severe complications or deaths occurred in this series. With our surveillance data, we cannot further describe the details of the clinical presentations, but in a recent report, PWH with CD4+T-cell count < 100 cells/μL and unsuppressed HIV viremia were identified at higher risk of severe disease, with risk for necrotizing skin and mucosal lesions, lung involvement with multifocal opacities (perivascular nodules), secondary bacterial infection, sepsis, and death [12]. Severity in mpox has mainly been reported in PWH with low CD4+T-cell count and has been proposed to be considered an AIDS-defining event. These individuals exhibited high rates of IRIS following ART initiation, including severe IRIS with a high mortality risk. However, the rate of coinfection with other opportunistic infections in these patients was high (26%), which could have confounded the observations.

Overall, a relatively high percentage of PWH developing mpox in our series were from Latin America (42%), including 100% of the immunosuppressed PWH, which may relate to ethnic disparities in HIV infection and inequities and barriers to HIV and STI prevention and mpox information and vaccination. Similarly, the US CDC has also recently reported a disproportionately higher incidence of mpox among Hispanic and Black/African American men in the US (31% and 33%, respectively) as well as a higher risk of death, indicating substantial health inequities and social risk factors in these groups [19,25]. Other studies have shown that assortative mixing in the selection of sexual partners may play a role in the increased incidence of HIV and other STIs in some clusters and might also have played a role in the racial disparities observed during the mpox epidemic [26]. It is crucial to ensure early and equal access to mpox and HIV prevention and treatment in all communities for continued control of both infections.

A strength of our study is the population-based dataset with all confirmed mpox cases reported in Catalonia, including individuals with HIV coinfection. We had access to the PISCIS cohort with data on CD4+T-cell count and HIV-RNA measurements close to the mpox diagnosis. In addition, we had access to high-quality comorbidity data in these patients and all notifiable STIs.

Our study has some limitations, including the surveillance nature of the data and not having detailed clinical data for the course of hospitalization regarding complications and treatment. Furthermore, our study includes a limited number of PWH with low CD4+T-cell count and, therefore, at risk for severe mpox disease, and it is therefore not possible to rule out type II errors in the analysis of this subpopulation.

In conclusion, unless immunosuppressed, PWH were not at increased risk of severe mpox or hospitalization compared to PWoH. PWH had more frequent fever and generalized umbilical or pustular exanthema. Mpox was associated with high rates of coinfection with HIV and other STIs and represented a marker of high-risk sexual behavior, supporting the importance of STIs and HIV testing in all mpox cases. Mpox diagnosis in HIV-negative individuals should trigger considerations to offer PrEP. We observed ethnicity disparities in mpox incidence during the outbreak, highlighting all communities’ need for equal access and utilization of healthcare services for continued efforts against mpox. Finally, dose-sparing smallpox vaccination was associated with a sharp and significant decline in mpox incidence in our study.

## Figures and Tables

**Figure 1 microorganisms-11-02701-f001:**
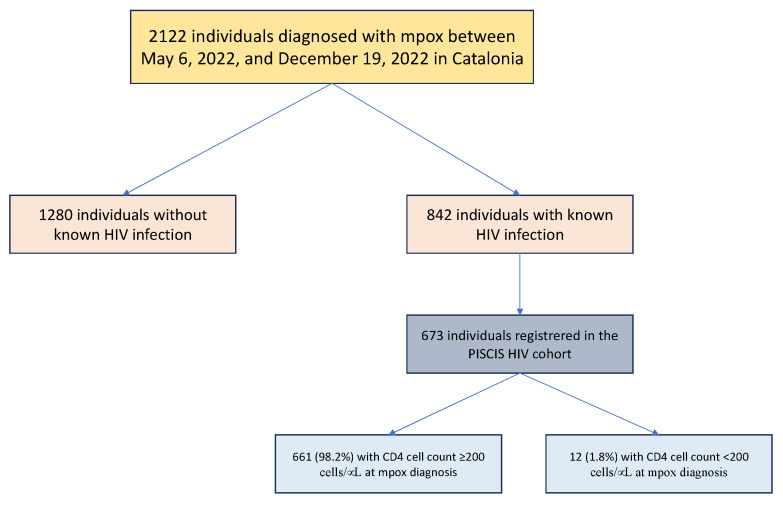
Flowchart.

**Figure 2 microorganisms-11-02701-f002:**
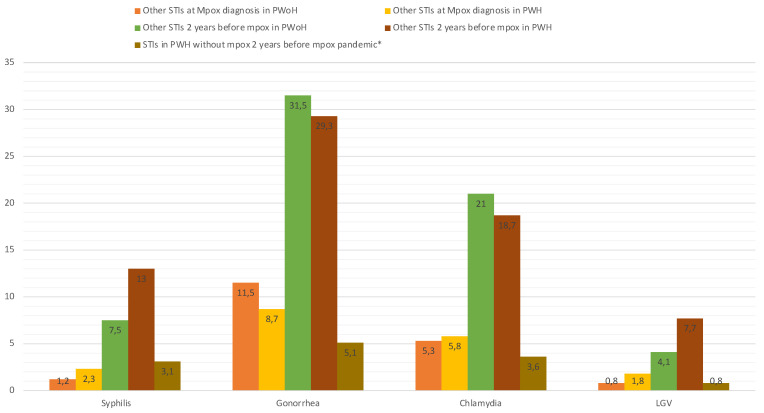
Sexually transmitted infections (STIs) in people with mpox (with and without HIV) and in people with HIV without mpox. Abbreviations: LGV, lymphogranuloma venereum; PWH, people with HIV; PWoH, people without HIV; STIs, sexually transmitted infections. * Between 1 May 2020 and 1 May 2022.

**Figure 3 microorganisms-11-02701-f003:**
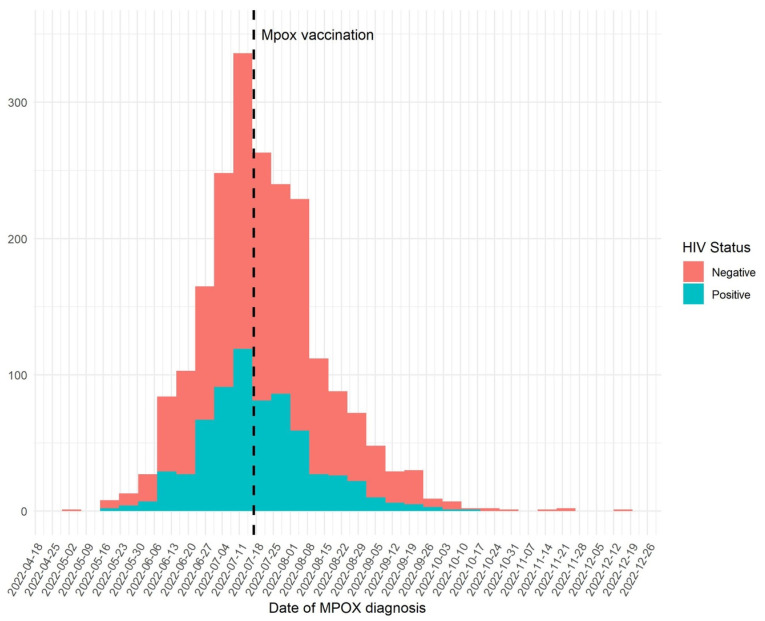
Epidemiological curves of weekly cumulative confirmed cases of mpox in Catalonia, from 6 May 2022 to 19 December 2022, in people with HIV (PWH) and people without HIV (PWoH).

**Table 1 microorganisms-11-02701-t001:** Clinical characteristics in people without HIV (PWoH) and people with HIV (PWH) infection at mpox diagnosis (*n* = 2122).

	Mpox in PWoH *n* = 1280	Mpox in PWH *n* = 842	*p*-Value
Age	36 (30–43)	40 (34–46)	<0.001
Male, *n* (%)	1229 (96.1)	840 (99.8)	<0.001
Origin, *n* (%)			
Spanish	756 (59.1)	438 (52.0)	0.001
European	192 (15.0)	113 (13.4)	0.31
Latin American	245 (19.1)	259 (30.8)	<0.001
Other	87 (6.8)	32 (3.8)	0.003
Asymptomatic, *n* (%)	62 (4.8)	20 (2.4)	0.004
Symptoms, *n* (%)			
Fever	641 (50.1)	467 (55.5)	0.015
Asthenia	394 (30.8)	243 (28.9)	0.35
Odynophagia	268 (20.9)	201 (23.9)	0.11
Myalgia	251 (19.6)	150 (17.8)	0.30
Headache	303 (23.7)	164 (19.5)	0.023
Generalized lymphadenopathy	153 (12.0)	94 (11.2)	0.58
Localized lymphadenopathy	530 (41.4)	346 (41.1)	0.89
Anogenital exanthema	727 (56.8)	472 (56.1)	0.74
Oro-facial exanthema	338 (26.4)	239 (28.4)	0.32
Exanthema in other localization	509 (39.8)	393 (46.7)	0.002
Type of exanthema			
Maculopapular	181 (14.1)	129 (15.3)	0.45
Vesicular	308 (24.1)	185 (22.0)	0.26
Pustular	237 (18.5)	194 (23.0)	0.011
Umbilicated	194 (15.2)	95 (11.3)	0.011
Crusts	127 (9.9)	83 (9.9)	0.96
Hemorrhagic	4 (0.3)	3 (0.4)	0.86
Complications *, *n* (%)	57 (4.5)	50 (5.9)	0.13
Skin bacterial infections	22 (1.7)	15 (1.8)	0.91
Localizations:			
Ano-genital	9 (34.6)	6 (42.9)	
Face	9 (34.6)	2 (14.3)	
Limbs	4 (15.4)	1 (7.1)	
Oral	4 (15.4)	5 (35.7)	
Corneal infections	5 (0.4)	3 (0.4)	0.9
Pneumonia	0	2 (0.2)	0.081
Proctitis	15 (1.2)	14 (1.7)	0.34
Hospitalization	22 (1.7)	22 (2.6)	0.16
ICU	1 (0.1)	0	0.42
Sepsis	0	0	
Encephalitis	0	0	
Death	0	0	
Other	3 (0.2)	0	
Long term complications	74 (9.3)	65 (12.3)	0.085

Abbreviations: ICU, intensive care unit, PWH, people with HIV; PWoH, people without HIV. * Complications were defined as having at least one of the following: skin bacterial infections, corneal infection, pneumonia, proctitis, and/or hospitalization.

**Table 2 microorganisms-11-02701-t002:** Clinical characteristics in people with HIV (PWH) in active follow-up in the PISCIS HIV cohort, according to CD4 cell count (*n* = 673).

	PWH with CD4 ≥ 200 Cells/μL at Mpox Diagnosis *n* = 661 (98.2)	PWH with CD4 < 200 Cells/μL at Mpox Diagnosis *n* = 12 (1.8)	*p*-Value
Demographics			
Male, *n* (%)	659 (99.7)	12 (99.7)	0.87
Age (years), median (IQR)	41 (35–46)	41 (32–44)	0.26
Birth area, *n* (%)			
Spain	268 (40.5)	0	
Eastern Europe	17 (2.6)	0	
Western Europe and Northern America	84 (12.7)	0	
Africa	11 (1.7)	0	
Latin America	271 (41.0)	12 (100)	<0.0001
Other	10 (1.5)	0	
Route of HIV transmission, *n* (%)			0.48
MSM	610 (95.5)	11 (91.7)	
Heterosexual men	13 (2.0)	1 (8.3)	
IDU	11 (1.7)	0	
Unknown	5 (0.8)	0	
Mpox			
Asymptomatic, *n* (%)	19 (2.9)	0	0.55
Symptoms, *n* (%)			
Fever	364 (55.1)	6 (50.0)	0.73
Asthenia	181 (27.4)	5 (41.7)	0.27
Odynophagia	144 (21.8)	3 (25.0)	0.79
Muscular pain	115 (17.4)	3 (25.0)	0.49
Headache	121 (18.3)	0	0.10
Generalized lymphadenopathy	73 (11.0)	2 (16.7)	0.54
Localized lymphadenopathy	264 (39.9)	4 (33.3)	0.64
Anogenital exanthema	350 (53.0)	6 (50.0)	0.84
Oro-facial exanthema	197 (29.8)	1 (8.3)	0.22
Generalized exanthema	295 (44.6)	10 (83.3)	0.008
Type of exanthema			
Maculopapular	97 (14.7)	3 (25.0)	0.32
Vesicular	130 (19.7)	3 (25.0)	0.65
Pustular	136 (20.6)	6 (50.0)	0.013
Umbilical	64 (9.7)	1 (8.3)	0.88
Crusts	52 (7.9)	2 (16.7)	0.26
Hemorrhagic	2 (0.30)	1 (8.3)	<0.0001
Complications, *n* (%)			
Bacterial infections	6 (0.9)	0	0.74
Cornea infections	3 (0.5)	0	0.82
Pneumonia	1 (0.2)	0	0.89
Proctitis	11 (1.7)	0	0.65
Hospitalization	13 (2.0)	2 (16.7)	0.001
HIV-related			
CD4 cell count at mpox, (cells/μL), median (IQR)	724 (540–912)	122 (72–176)	<0.0001
Viral load < 50 c/mL at mpox	553 (83.7)	8 (66.7)	0.12
Viral load < 200 c/mL at mpox	579 (87.6)	10 (83.3)	0.66
AIDS-defining event prior to mpox	51 (7.7)	4 (33.3)	0.001
Comorbidity			
Malignancy	38 (5.8)	3 (25.0)	0.006
Autoimmune disease	22 (3.3)	0	0.52
Inflammatory bowel disease	16 (2.4)	1 (8.3)	0.20
Multiple sclerosis	0	0	
History of psychiatric disease	207 (31.3)	3 (25.0)	0.64
Charlson comorbidity score at mpox,			0.23
0	523 (79.2)	10 (83.3)	
1	41 (6.2)	0	
2–3	87 (13.2)	1 (8.3)	
≥4	10 (1.5)	1 (8.3)	

Abbreviations: IDU, injection drug use; IQR, interquartile range; MSM, men who have sex with men; PWH, people with HIV; PWoH, people without HIV.

## Data Availability

The data collected for this study are available from the Centre for Epidemiological Studies of Sexually Transmitted Diseases and HIV/AIDS in Catalonia (CEEISCAT), the coordinating center of the PISCIS cohort study and from each of the collaborating hospitals upon request. Requests can be made via https://pisciscohort.org/contacte/. The study protocol, the statistical codebook, and codes for the analysis can be requested from RMI (raquel@bisaurin.org).

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
