# Peer review of "Mpox: Clinical Outcomes and Impact of Vaccination in People with and without HIV: A Population-Wide Study"

_microorganisms, 2023, doi:10.3390/microorganisms11112701_

Round 1

Reviewer 1 Report

Comments and Suggestions for Authors

Dear Editor , thank you for the opportunity to revise this manuscript by Raquel Martin-Iguacel et al. In this manuscript the authors present data about the clinical outcomes of mpox infection among people living with or without HIV. The authors found, as previously reported in other international and national cohorts, that only people with severe immunedepression were found to have more severe disease. Moreover, the authors present ecologic data on the effect of vaccination againts mpox.

I have some minor comments which I hope would help strenghtening the manuscript.

The manuscript is well written and worth sharing as it encompasses three different datasets (PISCIS, PADRIS and Mpox surveillance data) providing different information, including HIV status details and co-morbidities.

Moreover, the study includes over 2.000 individuals (40% with HIV), which help strenghtening previous findins of other smaller cohorts of PWH diagnosed with mpox.

Abstract. 

Line 39. please clarify "immunodepressed". Does this refer to CD4<200 as in the next sentence?

Results:

Lines 199-207: this paragraph is mainly related to vaccination, I would move this section below in the results, following all the discussion on the clinical characteristics among PWH

The authors might consider inclusing a comparison also between PWH with detectable versus undetectable HIVRNA

Can the authors clarify if all PWH were receiving ART? If not, how were mpox disease outcomes among people not receiving ART?

Can the authors clarify if there were any new HIV diagnosis concurrently to mpox diagnosis and related mpox outcomes? If yes, was ART introduced concurrently to mpox or following recovery?

Figure 2. Can the authors also provide in the Figure specific data on STIs at mpox diagnosis in PWoH and in PWH (e.g. two separate columns for PWH and PWoH)

I particularly praise Figure 3.

Author Response

Reviewers’ comments

Reply

Reviewer 1

Dear Editor, thank you for the opportunity to revise this manuscript by Raquel Martin-Iguacel et al. In this manuscript, the authors present data about the clinical outcomes of mpox infection among people living with or without HIV. The authors found, as previously reported in other international and national cohorts, that only people with severe immunodepression were found to have more severe disease. Moreover, the authors present ecologic data on the effect of vaccination against mpox.

I have some minor comments which I hope would help strengthening the manuscript.

The manuscript is well written and worth sharing as it encompasses three different datasets (PISCIS, PADRIS and Mpox surveillance data) providing different information, including HIV status details and co-morbidities.

Moreover, the study includes over 2.000 individuals (40% with HIV), which helps strengthening previous findings of other smaller cohorts of PWH diagnosed with mpox.

Abstract. 

Line 39. please clarify "immunodepressed". Does this refer to CD4<200 as in the next sentence?

We have changed the text, so it is more precise for the reader:

“PWH had no increased risk for complications, except those with CD4+T-cell<200 cells/µL.”

Results:

Lines 199-207: this paragraph is mainly related to vaccination, I would move this section below in the results, following all the discussion on the clinical characteristics among PWH

Thank you for the suggestion. As suggested, we have moved the paragraph to the end of the result section.

The authors might consider including a comparison also between PWH with detectable versus undetectable HIV-RNA.

Thank you for the comment.

We have included an additional table in the supplementary material (eTable 4) comparing the 561 PWH with HIV-RNA < 50 copies/ml versus the 47 PWH with HIV-RNA ≥50 c/ml as suggested by the reviewer (65 individuals were excluded from this analysis because they had unknown HIV-RNA at mpox).

We observed that the rate of hospitalization was also higher in the group of PWH with HIV-RNA ≥50 c/ml (8.5% versus 1.4%, p=0.001), but the median CD4 cell count in this group was high 463 (IQR 326-913). For this reason, we think the table in the manuscript comparing PWH with CD4 ≥200 cells/µL versus CD4 <200 cells/µL is more informative about the effect of immune suppression.

We have added to the text:

“Furthermore, we found that PWH with CD4+T-cell count <200 cells/mL were more commonly from Latin America (100% versus 41%, p<0.0001), had more generalized exanthema (83.3% versus 44.6%, p=0.008), had more frequent pustular and haemorrhagic exanthema (50.0% versus 20.6%, p=0.013 and 8.3% versus 0.3%, p<0.0001, respectively) and required more frequently hospitalization (16.7% versus 2%, p=0.001). Higher hospitalization rates were also observed in individuals with HIV-RNA ≥50 copies/ml compared to HIV-RNA< 50 copies/ml (eTable 4).”

Can the authors clarify if all PWH were receiving ART? If not, how were mpox disease outcomes among people not receiving ART?

The information on current ART in PISCIS (updated once yearly) does not necessarily reflect if the patient was taking the treatment at the moment of mpox diagnosis.

We believed that the information on HIV-RNA at mpox diagnosis is more reliable regarding whether the patient had been prescribed ART and was adherent to the treatment. Those who were not receiving ART would have many associated characteristics that we would not be able to capture from the database, including addictions or drug abuse, psychiatric diseases, homelessness, or worse social determinants of health.

Can the authors clarify if there were any new HIV diagnosis concurrently to mpox diagnosis and related mpox outcomes? If yes, was ART introduced concurrently to mpox or following recovery?

In our PISCIS database, no mpox cases were diagnosed concomitantly with HIV infection. The most recent HIV infection was recorded four months before mpox diagnosis. However, we cannot rule out concomitant HIV diagnosis at mpox diagnosis in the remaining 169 PWH who were not included in PISCIS, since only the HIV status was recorded in the standardized CRFs for these patients, with no date of HIV diagnosis.

We have added:

“There were no cases of concurrent mpox and HIV diagnosis in the PISCIS/PADRIS database. However, we cannot rule out concomitant new HIV diagnosis at mpox diagnosis in the remaining 169 PWH who were not included in PISCIS, since only the HIV status was recorded in the standardized CRFs for these patients. ”

Figure 2. Can the authors also provide in the Figure specific data on STIs at mpox diagnosis in PWoH and in PWH (e.g., two separate columns for PWH and PWoH)

We have modified the figure 2 as suggested. There are two columns for STIs at mpox diagnosis, one for PWoH and one for PWH.

All this data is also detailed in the supplementary material in eTable 3.

I particularly praise Figure 3.

Thank you

Reviewer 2 Report

Comments and Suggestions for Authors

This is a very interesting topic to me. But there are many problems should be addressed before publication.

Line 50: both mpox and monkeypox are listed as keywords.

Line 81-82, 100-101: reference should be given.

Line 199-200: any other comparative study to support the decline of confirmed cases is due to the vaccination? Is there other evidence to support the efficacy of smallpox vaccine on mpox?

Line 248-252: Were these control measures put forward to control the mpox outbreak in Catalonia? If so, how can you differentiate the contribution due to these measures or the smallpox vaccine?

Comments on the Quality of English Language

Line 156: aggregated? or cumulative may be better.

Line 174: fifty-nine percent and 21%. The format should be consistent.

Sometimes, there are blank lines following a paragraph in the main text such as line60, 80, 239, 283 but sometimes there are not.

Author Response

Reviewer 2

This is a very interesting topic to me. But there are many problems should be addressed before publication.

Line 50: both mpox and monkeypox are listed as keywords.

We appreciate very much the comment. We have now included only mpox.

We have also erased Monkeypox from the title.

Line 81-82, 100-101: reference should be given.

We have included the following references:

- Malone SM, et al. Review Safety and Efficacy of Post-Eradication Smallpox Vaccine as an Mpox Vaccine: A Systematic Review with Meta-Analysis. Intern J of Environmental Research and Public Health. 2023

+

- Titanji B. Effectiveness of Smallpox Vaccination to Prevent Mpox in Military Personnel. NEJM. 2023. DOI: 10.1056/NEJMc2300805

- Núm Disposición 2837 Del BOE Núm. 65 de 2015; 2015. Available from: https://www.boe.es/eli/es/o/2015/03/09/ssi445/dof/spa/pdf.

Line 199-200: any other comparative study to support the decline of confirmed cases is due to the vaccination? Is there other evidence to support the efficacy of smallpox vaccine on mpox?

-We have included:

- Malone SM, et al. Review Safety and Efficacy of Post-Eradication Smallpox Vaccine as an Mpox Vaccine: A Systematic Review with Meta-Analysis. Intern J of Environmental Research and Public Health. 2023

+

- Titanji B. Effectiveness of Smallpox Vaccination to Prevent Mpox in Military Personnel. NEJM. 2023. DOI: 10.1056/NEJMc2300805

+

Deputy, N.P.; Deckert, J.; Chard, A.N.; Sandberg, N.; Moulia, D.L.; Barkley, E.; Dalton, A.F.; Sweet, C.; Cohn, A.C.; Little, D.R.; et al. Vaccine Effectiveness of JYNNEOS against Mpox Disease in the United States. New England Journal of Medicine 2023, 388, 2434–2443, doi:10.1056/nejmoa2215201.

+

Dalton AF, Diallo AO, Chard AN, et al. Estimated Effectiveness of JYNNEOS Vaccine in Preventing Mpox: A Multijurisdictional Case-Control Study — United States, August 19, 2022–March 31, 2023. MMWR Morb Mortal Wkly Rep 2023;72:553–558. DOI: http://dx.doi.org/10.15585/mmwr.mm7220a3

as references to support the effect of vaccination

(ref. in line 261)

Line 248-252: Were these control measures put forward to control the mpox outbreak in Catalonia? If so, how can you differentiate the contribution due to these measures or the smallpox vaccine?

This is a very good question. Observational studies can only prove association but not causality. These measures were introduced early in Catalonia. We cannot differentiate the effect of these public health measures from the effect of smallpox vaccination other than the strong temporal association after the initiation of the vaccine campaign. We had discussed already this in lines 274-277:

“The introduction of the dose-sparing vaccine was associated with a clear decline in incidence in our population. However, other reasons, like the effect of other public health measures instigated, could have also played a role.”

We have added the following text, to clarify this issue further:

“Control measures such as contact tracing, information campaigns and vaccination were pivotal and were introduced early in Catalonia during the outbreak.”

Comments on the Quality of English Language

Line 156: aggregated? or cumulative may be better.

Thank you. We have replaced aggregated with cumulative, as suggested.

Line 174: fifty-nine percent and 21%. The format should be consistent.

As “Fifty-nine percent” is a numeric figure at the beginning of the sentence, we spelt it according to APA style guidelines.

Sometimes, there are blank lines following a paragraph in the main text such as line60, 80, 239, 283 but sometimes there are not.

We have revised the language and the formatting of the text.

Round 2

Reviewer 2 Report

Comments and Suggestions for Authors

My questions are addressed